# A model symbiosis reveals a role for sheathed-flagellum rotation in the release of immunogenic lipopolysaccharide

**Caitlin A Brennan[1], Jason R Hunt[2,3], Natacha Kremer[1], Benjamin C Krasity[1], Michael A Apicella[2,3], Margaret J McFall-Ngai[1], Edward G Ruby[1]***

[1]Department of Medical Microbiology and Immunology, University of Wisconsin-Madison, Madison, United States; [2]Department of Microbiology, The Carver College of Medicine, University of Iowa, Iowa City, United States; [3]Department of Internal Medicine, The Carver College of Medicine, University of Iowa, Iowa City, United States

**Abstract** Bacterial flagella mediate host–microbe interactions through tissue tropism during colonization, as well as by activating immune responses. The flagellar shaft of some bacteria, including several human pathogens, is encased in a membranous sheath of unknown function. While it has been hypothesized that the sheath may allow these bacteria to evade host responses to the immunogenic flagellin subunit, this unusual structural feature has remained an enigma. Here we demonstrate that the rotation of the sheathed flagellum in both the mutualist *Vibrio fischeri* and the pathogen *Vibrio cholerae* promotes release of a potent bacteria-derived immunogen, lipopolysaccharide, found in the flagellar sheath. We further present a new role for the flagellar sheath in triggering, rather than circumventing, host immune responses in the model squid-vibrio symbiosis. Such an observation not only has implications for the study of bacterial pathogens with sheathed flagella, but also raises important biophysical questions of sheathed-flagellum function.

***For correspondence:** egruby@wisc.edu

**Competing interests:** The authors declare that no competing interests exist.

## Introduction

Bacterial flagella are important virulence factors that facilitate tissue tropism in host–microbe interactions. Besides mediating such migration, flagella serve additional functions, such as adherence to or invasion of host cells (*Young et al., 2000*; *Hayashi et al., 2001*; *Giron et al., 2002*). However, in most vertebrate models of pathogenesis, it has been difficult to separate the distinct contribution of flagellar motility in bacterial migration from its other roles, such as immune-system activation (*Feuillet et al., 2006*; *Andersen-Nissen et al., 2007*). Bacterial flagella are highly conserved on a molecular level, but display diversity in morphology, number and subcellular localization between species. One such variation is the presence of an outer membrane-derived sheath (*Fuerst and Perry, 1988*; *Geis et al., 1993*; *Ferooz and Letesson, 2010*). While the flagella of most bacteria are unsheathed, such a structure surrounds the filament of several polarly flagellated bacteria, including several with a host-associated lifestyle. The biological function(s) of the sheath remains unknown, but it has been posited to allow these bacteria to evade innate immune recognition of the enclosed flagellins (*Gewirtz et al., 2004*; *Yoon and Mekalanos, 2008*). This hypothesized role for the flagellar sheath presents a conundrum: what is the effect of coating a rapidly spinning structure with lipopolysaccharide (LPS), another potent immunostimulatory molecule? As the tools with which to study the flagellar sheath are limited, we sought to use the model symbiosis between *Vibrio fischeri* and the Hawaiian bobtail squid, in which the events of initial colonization can be observed in real-time, to probe whether the sheathed flagellum plays a role in inducing a symbiont-specific host response.

**eLife digest** While a few of the bacteria that live in and on the bodies of humans and other animals are harmful and can cause disease, most others can offer benefits to their hosts. Many bacteria—including some important human pathogens—have tails called flagella that rotate to move the bacteria inside its host. However, the immune system can detect parts of these flagella and eliminate the pathogen.

Bacterial flagella are made from filaments of proteins, and some flagella are also enclosed by a sheath that is similar to the outer membrane that encloses certain bacteria. The function of this sheath is unclear, although some researchers have suggested that it might prevent the immune system from detecting the proteins in the flagellum. Now, by studying the interactions between the Hawaiian bobtail squid and a marine bacterium, Brennan et al. show that the sheath can actually alert the host that the bacteria are around.

The Hawaiian bobtail squid collects bioluminescent bacteria within a so-called 'light organ'. This organ undergoes a number of developmental changes to house the bacteria, and the squid then uses the light from the bacteria to mask its own shadow, which helps it to avoid being detected by predators. Brennan et al. compared how wild-type bacteria and mutant bacteria that either had no flagella, or had flagella that did not rotate, interacted with young squid. Only bacteria with working flagella were able to trigger the normal development of the squid's light organ, which suggests that the rotating flagella are releasing the signal that tells the squid that the beneficial bacteria are present.

Brennan et al. demonstrated that the rotation of sheathed flagella led to the release of a molecule called lipopolysaccharide. This molecule is known to activate the immune system in animals, and it is one of the bacterial signals that the squid responds to. Moreover, when the flagella of other bacteria with sheaths—such as those that cause cholera—are rotating, there is also an increase in the release of lipopolysaccharide. However, rotation of the flagella of bacteria without sheaths has no such effect. The next challenge will be to test the importance of this release of lipopolysaccharide from rotating flagella on the outcome of bacterial diseases of humans and other animals.

*V. fischeri* is a gram-negative, bioluminescent bacterium that is found both in seawater and as a beneficial symbiont colonizing the light-emitting organs of certain fishes and squids. In the mutualism with the Hawaiian bobtail squid, *Euprymna scolopes*, symbiosis begins when a newly hatched squid specifically recruits *V. fischeri* cells from ambient seawater. The bacteria first aggregate on ciliated appendages of the light organ, and then migrate through pores and down ducts, ultimately reaching internal crypts (*Figure 1A*) where the symbiont population is housed (*Nyholm and McFall-Ngai, 2004*). Successful colonization of the crypts ultimately requires flagellar motility, as amotile mutants are unable to colonize the light-organ to luminous levels (*Graf et al., 1994*; *Brennan et al., 2013*). However, the symbionts are sensed by host cells even before they reach the crypts. Indeed, as few as 3–5 bacteria interacting with the ciliated cells of the surface appendages signal both organ-wide changes in host transcription and immune-cell (hemocyte) trafficking (*Koropatnick et al., 2007*; *Kremer et al., 2013*). Colonizing *V. fischeri* cells also induce an extensive morphogenesis of the light organ, beginning with the programmed cell death (apoptosis) of the ciliated appendages that begins within 6 hr of initial exposure to the symbiont, and continues until their complete regression (*Montgomery and McFall-Ngai, 1994*; *Foster et al., 2000*).

## Results

To address whether the sheathed flagellum mediates symbiont recognition, we first compared the appearance of apoptosis within the host's ciliated epithelium (*Figure 1A*) in response to inoculation with either wild-type *V. fischeri*, or one of two amotile mutants: *motB1* (*Brennan et al., 2013*), which bears normal flagella that do not rotate (*Figure 1—figure supplements 1 and 2*); and *flrA*, which is both amotile and aflagellate (*Millikan and Ruby, 2003*; *Brennan et al., 2013*; *Figure 1—figure supplements 1 and 3*). Such amotile mutants do not successfully colonize the light-organ deep crypts (*Graf et al., 1994*). However, as this apoptosis proceeds over the first few days of symbiotic colonization,

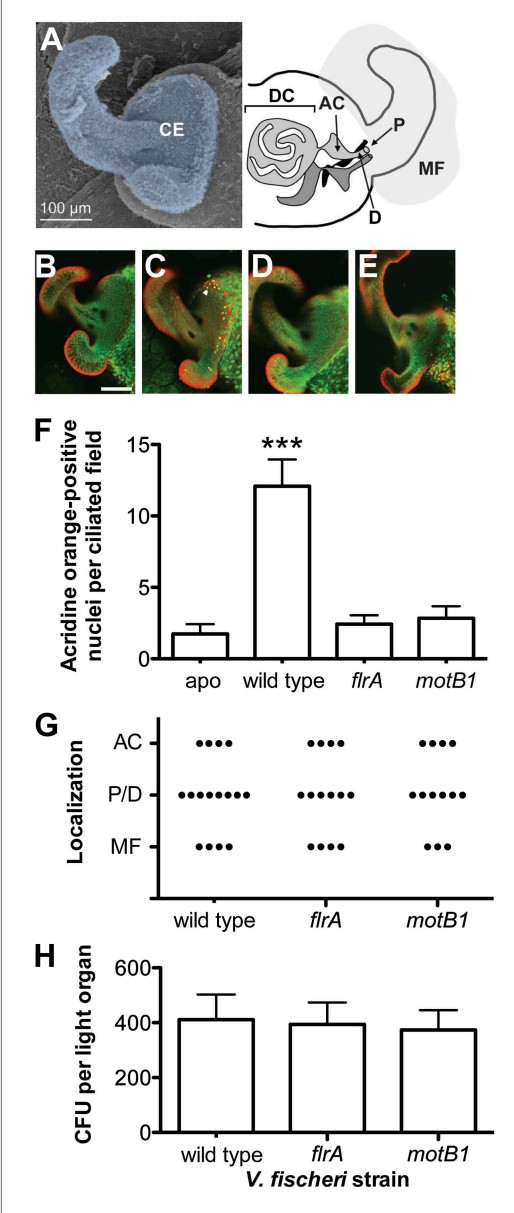

**Figure 1.** Motility mutants of *V. fischeri* do not induce early-stage apoptosis. (**A**) Morphology of the juvenile light organ, highlighting the external ciliated epithelium (CE; false-colored blue) in a scanning electron micrograph (left), and the internal features with which the bacteria interact during initiation of symbiotic colonization in a schematic diagram (right). DC, deep crypts; AC, antechamber; D, duct; P, pore; and MF, mucus field. (**B–E**) Representative laser-scanning confocal microscopy (LSCM) images of acridine orange (AO)-stained juvenile light organs, after exposure to either: no *V. fischeri* (**B**, apo); wild type (**C**); *flrA* (**D**); or *motB1* (**E**), as described in the 'Materials and methods'. White arrowhead in (**C**) indicates one of the numerous AO-positive nuclei (yellow). Scale bar in (**B**) represents 100 μm for all the images. (**F**) Counts of AO-positive

we assessed this phenotype an early time-point (10 hr post-inoculation) to optimize apoptopic induction by *V. fischeri* while selecting conditions under which amotile mutants would not yet exhibit a colonization defect. As previously reported (**Montgomery and McFall-Ngai, 1994**), squid that were not exposed to *V. fischeri* (i.e., aposymbiotic, or 'apo') exhibited much lower levels of apoptosis (**Figure 1B,F**) than squid exposed to wild-type *V. fischeri* (**Figure 1C,F**). In contrast, exposure to either the *motB1* or the *flrA* strain did not induce early-stage apoptosis above the background levels (**Figure 1D–F**). Because the actual location at which amotile mutants are arrested in symbiotic initiation has not been defined, we confirmed that both the location and number (**Figure 1G,H**) of bacteria at this early stage were indistinguishable in squid exposed to wild-type, *flrA* or *motB1* cells. These data show that flagellar motility plays a role in inducing a specific host immune response that is independent of its later role in migration into the light-organ crypts (**Brennan et al., 2013**), suggesting that biosynthesis of the flagellum, and its subsequent functions, may mediate multiple aspects of symbiotic initiation.

Reports of flagellar-mediated activation of immunity have thus far been limited to responses to the filament protein, flagellin (**Hayashi et al., 2001**; **Franchi et al., 2006**). Despite the flagellar sheath, flagellin monomers are a prevalent component of the secretome of both *V. cholerae* (**Xicohtencatl-Cortes et al., 2006**) and *V. fischeri* (unpublished data). However, the failure of the fully flagellated *motB1* mutant to induce normal apoptosis demonstrated that the expression of immunogenic flagellin is insufficient to activate apoptosis in the light organ and, instead, suggests that apoptosis is driven by flagellar rotation itself. Previous work (**Foster et al., 2000**) has shown that apoptosis of the light-organ ciliated epithelium is induced specifically by the lipid-A component of LPS (**Figure 1—figure supplements 4 and 5**); indeed, LPS is a common activator of innate-immune defenses across the animal kingdom (**Choi et al., 1998**; **Manna and Aggarwal, 1999**). However, no effect of flagellar motility on LPS presentation has been reported previously. We posited that, as sheath production and flagellar biogenesis may not be tightly coordinated (**Richardson et al., 1990**; **Josenhans et al., 1995**; **Ferooz and Letesson, 2010**), a sheathed flagellum might shed LPS due to a rotation-weakened association of the sheath with the spinning flagellar filament. This hypothesis is supported by negative-stain transmission electron microscopy (TEM) of

*Figure 1. Continued*

nuclei in the light-organ ciliated epithelium of squid exposed to either the indicated *V. fischeri* strains, or no *V. fischeri* (apo). (***), p<0.001 by Kruskal–Wallis Analysis of Variance (ANOVA), followed by Dunn's Multiple Comparison test. The absence of apoptosis in the aposymbiotic squid confirms that the experimental addition of peptidoglycan (PGN) ('Materials and methods') is insufficient to induce AO staining. (**G**) Bacterial localization under conditions used for measurement of early-stage apoptosis (panel **F**). Localization was determined by examining squid exposed to the indicated strains, genetically labeled to express GFP, using LSCM. Each point represents the bacterial localization in a single light-organ lobe. (**H**) Level of colonization (colony-forming units, CFU) under conditions used for measurement of early-stage apoptosis (panel **F**). Light organs were dissected from anesthetized and pithed squid, and then homogenized and plated to determine symbiont number.

The following figure supplements are available for figure 1:

**Figure supplement 1**. Soft-agar motility of wild-type, *flrA* and *motB1* strains.

**Figure supplement 2**. Negative-stain transmission electron micrograph of a representative flagellated *motB1* cell.

**Figure supplement 3**. Negative-stain transmission electron micrograph of a representative aflagellate *flrA* cell.

**Figure supplement 4**. Representative LSCM image of an AO-stained light-organ lobe isolated from a squid treated with exogenous *V. fischeri* lipid A.

**Figure supplement 5**. Induction of early-stage apoptosis in response to exogenous lipid A.

sheathed flagella, which has revealed vesicle-like structures (*Figure 2A*), often at the distal tip of the flagellum, and ~10–80 nm in diameter (*Geis et al., 1993*; *Millikan and Ruby, 2004*; *Ferooz and Letesson, 2010*). Such images are indicative of weaknesses between the flagellar sheath and shaft, which may be analogous to the foci of disassociation between the outer membrane and peptidoglycan required for the formation of cell-derived outer-membrane vesicles (OMVs) (*McBroom and Kuehn, 2007*). Thus, rotation of the flagellum would further destabilize these interactions, likely releasing LPS and activating an LPS-dependent immune response, rather than simply shielding against a host response.

We determined whether the *motB1* and *flrA* mutants released less of this signal molecule than wild-type *V. fischeri* using the *Limulus* amoebocyte lysate (LAL) assay to quantify the amount of reactogenic LPS in cell-free supernatants prepared from exponentially growing cultures (*Figure 2B*). While all strains had similar growth characteristics (*Figure 2—figure supplement 1*), we observed a significant reduction in shed LPS by the *motB1* mutant, whose flagella do not rotate, and an even greater reduction by the aflagellate *flrA* mutant. To confirm that these strains actually released less (and not just a less reactive) LPS, we directly measured the total LPS shed from these strains by purifying the lipid fraction from the cell-free culture supernatants, and visualizing the LPS by quantitative SDS-PAGE (*Figure 2—figure supplement 2*). This analysis revealed a substantial reduction in total LPS released by both the *motB1* and *flrA* strains (*Figure 2C*), consistent with the data from the LAL assay (*Figure 2B*).

To further test our hypothesis, we next examined whether (i) genetic complementation of the *motB1* mutant would restore normal levels of LPS shedding; (ii) disruption of flagellar rotation by a *motB1*-independent method would similarly lessen the LPS observed in culture supernatant; and, (iii) reduced LPS shedding would be observed when flagellar rotation is disrupted in other sheathed bacteria as well. As anticipated, expression of *motB1 in trans* restored both soft-agar motility and reactogenic LPS levels in cell-free culture supernatants (*Figure 3*, *Figure 3—figure supplement 1*). In addition, cell-free culture supernatants from a *V. fischeri motX* mutant, which lacks a different component of the flagellar motor than *motB1*, yet is also flagellated but amotile (*Figure 4—figure supplements 1 and 2*), had similarly reduced levels of reactogenic LPS (*Figure 4*). Taken together, these data support the notion that flagellar rotation promotes LPS release by *V. fischeri* cells.

Finally, we asked whether flagellar rotation-mediated LPS release is characteristic of other bacteria with sheathed flagella by comparing the amount of LPS shed by wild-type *Vibrio cholerae* O395N1 and its amotile *motAB*-deletion derivative. As in *V. fischeri*, disruption of flagellar rotation in *V. cholerae* significantly reduced LPS release (*Figure 5A*), supporting the conclusion that this mechanism of LPS release is not unique to *V. fischeri*, but is conserved among bacteria with sheathed flagella. In contrast, a similar analysis of an *Escherichia coli* mutant unable to rotate its (unsheathed) flagella revealed no alteration in levels of released LPS (*Figure 5B*), further indicating that this behavior may be specific to bacteria that sheath their flagella. Nevertheless, it remains possible that rotation-mediated LPS

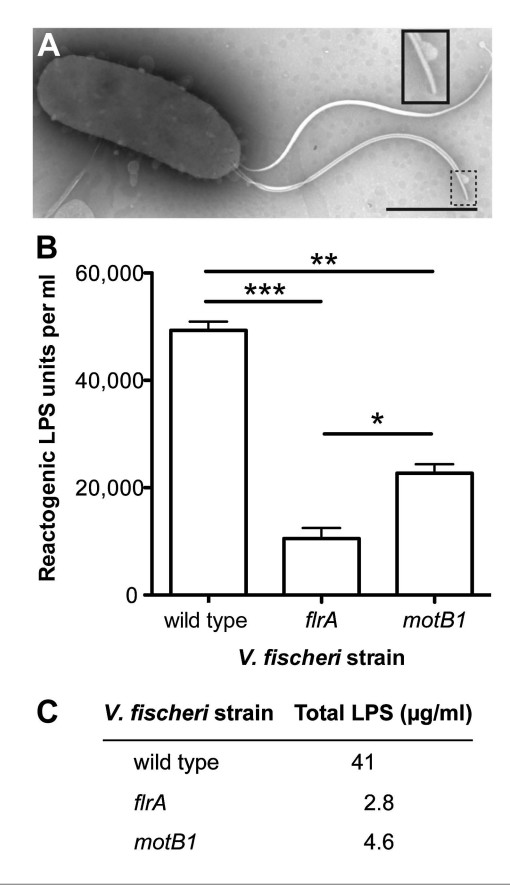

**Figure 2**. Motility mutants of *V. fischeri* release less LPS into culture supernatants. (**A**) Negative-stain TEM of a wild-type *V. fischeri* cell. Dashed box, shown larger within the solid box, highlights the distal tip of two flagella, one of which displays dissociation of the sheath from the filament in the form of a membrane vesicle-like structure. Scale bar indicates 1 µm. (**B**) Reactogenic LPS in cell-free supernatants of *V. fischeri* mid-log ($OD_{600} \approx 0.5$) cultures grown in seawater tryptone (SWT) medium at 28°C was measured by LAL assay. (*), $p<0.05$; (**), $p<0.01$; and (***), $p<0.001$, as analyzed by one-way repeated measures ANOVA, with a posthoc Bonferroni correction. (**C**) Total LPS levels in cell-free supernatants from mid-log ($OD_{600} \approx 0.5$) cultures grown in SWT at 28°C for indicated strains were determined by quantitative SDS-PAGE analysis (detailed in *Figure 2—figure supplement 1*).

The following figure supplements are available for figure 2:

**Figure supplement 1**. Growth of *V. fischeri* wild-type, *motB1* and *flrA* strains.

**Figure supplement 2**. Quantification of total LPS in cell-free supernatants by SDS-PAGE analysis.

release still occurs in other non-sheathed bacteria, or, perhaps, under other growth conditions.

## Discussion

Using the model squid-vibrio symbiosis as a tool to probe the effect of the enigmatic flagellar sheath on host recognition, we uncover a previously unrecognized role for the rotation of sheathed flagella in mediating the release of the immunogenic molecule LPS in both *V. fischeri* and its pathogenic congener *V. cholerae*. The flagellar sheath has proved problematic to study; indeed, to our knowledge, there are no mutations that result in the loss of a sheath in otherwise sheathed flagella. Intriguingly, a few *Vibrio* species, including the human pathogen *Vibrio parahaemolyticus*, encode two independent flagellar structures: one set producing sheathed, the other unsheathed flagella (*McCarter, 1999*). However, pleiotropic effects of mutations in these distinct pathways as well as other technical limitations make a direct comparison of unsheathed and sheathed flagella within this species difficult to interpret (L McCarter, personal communication). Thus, the functional role of the sheath, even as it relates to bacterial motility, remains poorly characterized, and observation of this new phenomenon for the flagellar sheath furthers our understanding of this enigmatic structure and its contribution to bacterial virulence. As this behavior is difficult to probe experimentally, novel approaches, such as the modeling of the biophysical properties underlying lipid-membrane behavior at a sub-micron scale, are needed to inform our understanding of the interactions between the sheath and filament along the length of the flagellum, as well as the physical forces at play between a rotating flagellar sheath and the contiguous, but static, cellular outer membrane. While vesicle-like structures have only been reported at the flagellar tip of *V. fischeri* (*Millikan and Ruby, 2004*), another possible region from which LPS may be shed during rotation is the base of the flagellum where the basal body perturbs the outer membrane. Interestingly, cryotomographic imaging has revealed unusual features in the *Vibrio* flagellar basal body that may help stabilize this interaction (*Chen et al., 2011*).

Additionally, our results show that flagellar rotation in *V. fischeri* promotes release of the immunogenic molecule LPS that, during initiation of symbiosis, is required for normal induction of apoptotic tissue development (*Foster et al., 2000*). Further, by activating the squid's developmental program in response to symbiont exposure, the flagellar sheath may serve to promote recognition by the host immune system, suggesting the

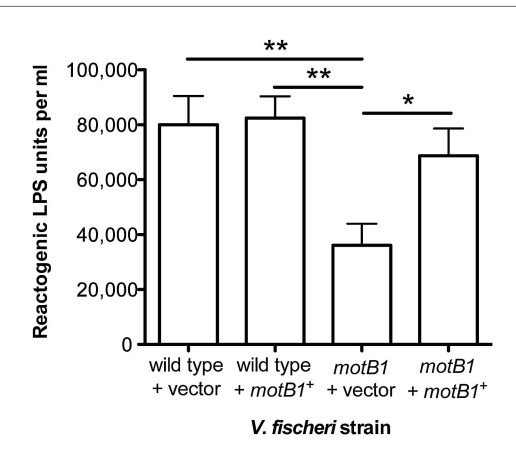

**Figure 3**. Genetic complementation restores supernatant LPS levels to the *motB1* mutant strain. Levels of reactogenic LPS in mid-log culture supernatants ($OD_{600} \approx 0.5$) of indicated strains, as measured using the LAL assay. (*), $p<0.05$, and (**), $p<0.01$, as analyzed by one-way repeated measures ANOVA with a posthoc Bonferroni correction.
The following figure supplements are available for figure 3:

**Figure supplement 1**. Soft-agar motility of genetically complemented *motB1* strains.

context-dependent nature (mutualism or pathogenesis) of this behavior. Perhaps masking one immunostimulatory molecule (e.g., flagellin) with another (e.g., LPS) can serve to modulate, rather than simply avoid (*Gewirtz et al., 2004*; *Yoon and Mekalanos, 2008*), detection by the host immune system.

Surprisingly, although both the *flrA* and *motB1* mutants still release LPS in culture, albeit at a reduced level, we observed complete attenuation of apoptotic induction, suggesting that (i) the threshold of LPS sensed by host cells is exquisitely calibrated to the amount of LPS shed by *V. fischeri*; (ii) because *V. fischeri* cells are not dividing during symbiotic initiation, flagellar-mediated LPS release is the predominant source of LPS to which host cells are exposed; and/or (iii) the LPS released by wild-type *V. fischeri* is somehow presented in a more immunogenically active form than that released by either the *flrA* or *motB1* mutant. This last point emphasizes the microheterogeneous chemical structures of LPS made by bacteria like *V. fischeri* (*Phillips et al., 2011*). This heterogeneity, as well as LPS's insolubility in aqueous solution and the toxicity of spent medium to juvenile squid, makes directly comparing the chemical concentration or specific activity of LPS between culture and symbiotic settings problematic. In this symbiosis, the LPS is produced by a few non-growing bacterial cells at a tissue interface, in high flow-field conditions that further confound quantification.

Finally, if LPS is shed from the flagellar tip as vesicles, other molecules such as flagellin might be enclosed, and these components may act synergistically to induce host responses (although flagellin has not yet been characterized as a morphogen in this association). Alternatively, LPS released from either the tip or base of the flagellum might form vesicles with a smaller diameter than the cell-derived OMVs, thereby enabling tighter interaction with the host-cell surfaces. Further work will build upon these discoveries and clarify the mechanisms underlying signaling during initiation of the squid-vibrio symbiosis; such work includes characterizing the composition of LPS released by *V. fischeri* during flagellar rotation, identifying the squid receptors that recognize *V. fischeri* LPS (*Goodson et al., 2005*; *Krasity et al., 2011*), and determining any role flagellin may have in activating host responses.

While thus far demonstrated only in *V. fischeri* and *V. cholerae*, the phenomenon of flagellar-mediated LPS release may be conserved across other animal-associated bacteria with sheathed flagella. The human pathogens *Brucella melitensis* and *Helicobacter pylori*, whose sheathed flagellum and/or flagellar motility serve as important virulence factors (*Ottemann and Lowenthal, 2002*; *Petersen et al., 2011*), may similarly release the immunogenic molecule LPS during flagellar rotation and, thereby, regulate immune recognition during infection. However, the complexity of mammalian models of host–microbe interactions can limit efforts to separate the various contributions of flagellar motility, and highlights the role of invertebrate model systems in revealing the underlying principles that govern shared mechanisms of host–microbe interactions.

## Materials and methods

### Bacterial strains and growth conditions

*V. fischeri* strains used in this study include: the wild-type *E. scolopes* light-organ isolate, ES114 (*Boettcher and Ruby, 1990*); *flrA*::Tn*erm*, MB21407 (*Brennan et al., 2013*); *motB1*::Tn*erm*, MB06357 (*Brennan et al., 2013*); and *motX*::Tn*erm*, MB12561 (*Brennan et al., 2013*). Strains were grown at 28°C in either Luria–Bertani salt (LBS) medium (per liter, 10 g Bacto-tryptone, 5 g yeast extract, 20 g NaCl, and 50 ml of 1 M Tris–HCl buffer, pH 7.5, in deionized water) for overnight growth, or seawater

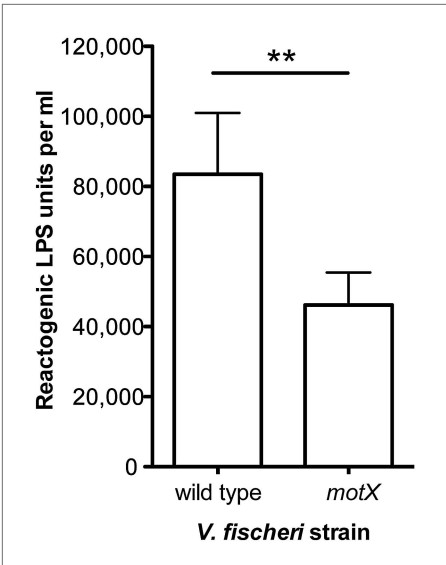

**Figure 4**. Disruption of *motX* in *V. fischeri* reduces supernatant LPS levels. Levels of reactogenic LPS, as measured using the LAL assay, in cell-free supernatants of mid-log cultures (OD$_{600}$ ≈ 0.5) of *V. fischeri* wild-type and *motX* strains. (\*\*), p<0.01 as determined by paired Student's *t* test.

The following figure supplements are available for figure 4:

**Figure supplement 1**. Soft-agar motility of *V. fischeri* *motX* mutant.

**Figure supplement 2**. Negative-stain transmission electron micrograph of a representative *motX* cell.

tryptone (SWT) medium (per liter, 5 g Bacto-tryptone, 3 g yeast extract, 3 ml glycerol, 700 ml Instant Ocean [Aquarium Systems, Inc., Mentor, OH] at a salinity of 33–35 ppt, and 300 ml distilled water) for experimental use. *V. cholerae* strains O395N1 and its ΔpomAB derivative (*Gosink and Hase, 2000*) (referred to as *motAB* within this work, for consistency with the *V. fischeri* and *E. coli* nomenclature) were grown at 37°C in either SWT for LPS measurements, or Luria-Bertani medium (per liter, 10 g Bacto-tryptone, 5 g yeast extract, and 10 g NaCl) for overnight growth. *E. coli* strains RP437 and its Δ*motAB* derivative, RP6894 (*Zhou et al., 1998*), were grown at 30°C in tryptone medium (per liter, 10 g Bacto-tryptone and 5 g NaCl). When appropriate, antibiotics were added to media at the following concentrations: erythromycin, 5 µg/ml; and chloramphenicol, 2.5 µg/ml. Growth media were solidified with 1.5% agar unless otherwise indicated.

For growth curves, overnight cultures were used to inoculate 100 µl volumes of SWT, arrayed in a 96-well plate, and a plate-reader was used to measure optical density (OD) at 600 nm over 9 hr of growth at 28°C with shaking. Data are presented as the mean, and standard error of the mean, for three biological replicates.

## Assessment of motility and flagellar structure

For soft-agar motility assays, cells were grown to OD$_{600}$ ≈ 0.3–0.4. Cultures were then normalized to an OD$_{600}$ of 0.3, and 2 µl of each strain were inoculated into plates containing SWT supplemented with 0.3% agar. Plates were grown at 28°C for 8–11 hr. The diameter of migration by the cells is a measure of their level of flagellar motility.

For examination of flagellar structures, cells were grown in SWT broth with shaking at 28°C to an OD$_{600}$ of ~0.3, applied to Pioloform-coated copper grids (Ted Pella Co., Tustin, CA) for 1 min, washed with sterile water for 30 s, and negatively stained for 1 min with NanoW (Nanoprobes, Yaphank, NY). Grids were immediately examined using a Philips CM120 transmission electron microscope (University of Wisconsin Medical School Electron Microscope Facility, Madison, WI).

## Construction of the *motB1* complementation strain

The *motB1* open reading frame was amplified by PCR using primer pair motB1_compF (5'-GCTCTAGAC TAACACACAGGAAACAGCTATGGAAGATGAAAACGACTGCA-3'), which introduces an XbaI restriction enzyme site and a ribosomal binding site, and motB1_compR (5' GCGGTACCGACCTAATCTAA GGCGCA-3'), which introduces a KpnI restriction enzyme site. The resulting product was cloned into the XbaI/KpnI-digested fragment of pVSV105 (*Dunn et al., 2006*). Both the complementation construct (pCAB98) and vector control were conjugated into wild-type *V. fischeri* and into the *motB1* mutant using standard techniques (*Stabb and Ruby, 2002*).

## Measurement of reactogenic LPS using a chromogenic *Limulus* amoebocyte lysate (LAL) assay

*V. fischeri* cultures were prepared by growth in SWT broth with shaking to mid-log phase (OD$_{600}$ ≈ 0.5). Cells were removed by two successive rounds of centrifugation, and the resultant supernatant was passed through a 0.22-µm pore-sized filter, yielding a purified, cell-free supernatant. Cell-free supernatants were serially diluted in pyrogen-free water, and reactogenic LPS was detected from these samples

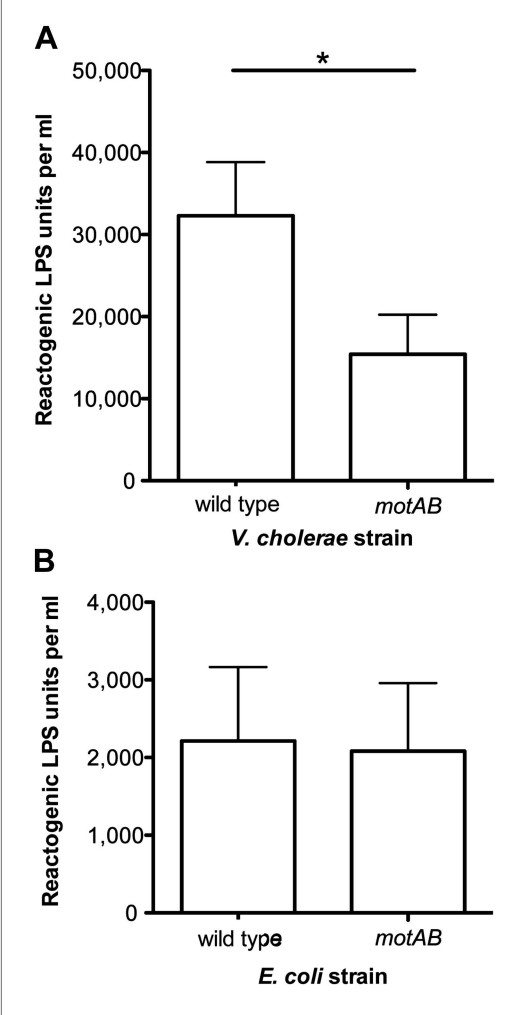

**Figure 5**. Loss of flagellar rotation reduces LPS release by sheathed *V. cholerae*, but not unsheathed *E. coli*. (**A**) Reactogenic LPS released into mid-log (OD$_{600}$ ≈ 0.5) culture supernatants by wild-type *V. cholerae* O395N1 and its *motAB* (i.e., *pomAB)* derivative, grown in SWT at 37°C, was measured by the LAL assay. (\*), p<0.05 by paired Student's *t* test. (**B**) Reactogenic LPS in mid-log (OD$_{600}$ ≈ 0.5) culture supernatants of wild-type *E. coli* and its *motAB* derivative grown in tryptone medium at 30°C, as measured using the LAL assay.

using the ToxinSensor Chromogenic LAL Endotoxin Assay Kit (Genscript, Piscataway, NJ), per manufacturer's instruction, except in half–volume reactions. Reactogenic LPS units for each sample were normalized to the OD$_{600}$ of the culture. The LPS in uninoculated media, serially diluted in pyrogen-free water, was determined to be fewer than 50 reactogenic LPS units per ml. Data are presented as the mean, and standard error of the mean, for at least three sets of biological replicates. Statistical analyses were performed with a repeated-measures ANOVA, or paired Student's *t* test when appropriate, to normalize for variability in the basal levels of LPS released by wild-type cells across different experiments.

## Measurement of total LPS by quantitative, silver-stained, SDS-PAGE analysis

To remove salts from 50 ml of purified cell-free supernatants of *V. fischeri* (described above in 'Measurement of reactogenic LPS using a chromogenic *Limulus* amoebocyte lysate (LAL) assay'), samples were dialyzed against four liters of distilled, deionized H$_2$O using 3500 MWCO SnakeSkin dialysis tubing (ThermoScientific, Rockford, IL) and five water changes. The dialyzed samples were lyophilized and resuspended in 10 ml DNaseI buffer (10 mM Tris–HCl, pH 7.6, 2.5 mM MgCl$_2$, 0.5 mM CaCl$_2$) and treated with 1 mg/ml DNaseI/RNaseA (Roche, Indianapolis, IN) overnight at 37°C. Following nuclease treatment, an equal volume of 95% phenol at 65°C was added. Samples were vortexed and incubated at 65°C for 30 min, then cooled on ice and centrifuged at 3000×*g* for 10 min at 4°C. The aqueous layer was collected and the phenol layer was back extracted with an equal volume of deionized water pre-warmed to 65°C. The aqueous layers were combined and residual phenol was removed by addition of one-tenth volume of 0.3 M sodium acetate, and precipitation three times with 3 volumes of absolute ethanol. The resulting pellets were resuspended in HPLC-grade water, frozen, and again lyophilized.

Total purified LPS samples from each 50 ml of culture were measured for dry weight, resuspended to a final concentration of 25 μg/μl, and analyzed by SDS-polyacrylamide gel electrophoresis (PAGE) using NuPAGE pre-cast 4–12% Bis-Tris polyacrylamide gels (Novex, Grand Island, NY). The gels were loaded with 5 μl from each preparation, as well as known masses of purified *V. fischeri* LPS to produce a standard curve. Following electrophoresis, gels were analyzed by silver staining and densitometry using ImageJ. A standard curve was generated based on the band intensities of the purified *V. fischeri* LPS fractions, and the concentrations of LPS in the culture supernatants were estimated from this standard curve.

## Squid colonization experiments

Freshly hatched juvenile squid were exposed to 100 μg of lysozyme-treated peptidoglycan per ml of filter-sterilized Instant Ocean at 32–36 ppt (PGN-FSIO) for 2 hr to induce mucus shedding (*Nyholm*

*et al., 2002*). In addition to enhancing the solubility of the peptidoglycan, this lysozyme treatment can result in smaller, more immunogenic breakdown products (like TCT). As indicated in individual experiments, either *V. fischeri*, at a final concentration of ~5000 colony-forming units (CFU) per ml, or 1 µg/ml *V. fischeri* lipid A were added to the PGN-FSIO.

After 10 hr, squid were exposed to 0.001% acridine orange (AO) for 5 min, washed in filter-sterilized Instant Ocean (FSIO), and anesthetized in 2% ethanol in FSIO. Squid were then ventrally dissected to expose the light organ, and examined by laser-scanning confocal microscopy (LSCM). AO-positive nuclei in the ciliated epithelial field were counted for one light-organ lobe per squid as a measure of cell-death induction (*Delic et al., 1991*; *Foster et al., 2000*). Data are presented as the mean, and standard error of the mean, for 32–38 squid per condition.

For experiments in which symbiont number was determined, squid were colonized as above and, at 10 hr post-inoculation, rinsed three times in FSIO, anesthetized in 2% ethanol in FSIO, and ventrally dissected to remove the light organ. Individual light organs were then homogenized, and each homogenate was diluted and spread onto LBS agar for CFU. Data are presented as the mean, and standard error of the mean, for 36–37 squid per condition.

To visualize bacterial localization, squid were similarly colonized with bacterial strains carrying pVSV102, which harbors a green-fluorescent protein (GFP)-encoding gene under constitutive expression (*Dunn et al., 2006*), using the conditions described for assaying early-stage apoptosis. After 9.5 hr in PGN-FSIO containing the *V. fischeri* inoculum, animals were exposed to Cell Tracker Orange CMRA (Invitrogen Molecular Probes, Carlsbad, CA) for 30 min to label host tissue. Squid were then anesthetized in 2% ethanol in FSIO, pithed, and examined by LSCM as described previously (*Nyholm et al., 2000*; *Sycuro et al., 2006*). Bacterial localization was determined by the presence of GFP-fluorescing cells, relative to notable features of the juvenile light organ (*Figure 1A*). Data indicate the location of the bacterial aggregate that has progressed the furthest in a single light-organ lobe, for 13–15 squid per condition. Thus, the decreased induction of apoptosis by these mutants does not result simply from a differential arrest earlier in migration into the light organ crypts.

Scanning electron microscopy was performed on freshly hatched squid, as previously described (*Koch et al., 2013*).

## Acknowledgements

We thank E Koch (University of Wisconsin–Madison) for graciously providing the scanning electron micrograph of a juvenile squid, C Häse (Oregon State University) for supplying *V. cholerae* strains, JS Parkinson (University of Utah) for providing *E. coli* strains, and RA Welch (University of Wisconsin–Madison) for sharing research space. We thank Morgan Beeby (Imperial College London) and Rob Phillips (Caltech) for helpful discussions. This work was supported by NIH grants RR12294/OD11024 (to EGR and MJM-N) and AI50661 (to MJM-N). CAB was supported by an NIH Molecular Biosciences Training Grant (T32 GM07215), and an NIH Microbes in Health and Disease Training Grant (T32 AI055397), to the University of Wisconsin–Madison.

## Additional information

### Funding

| Funder | Grant reference number | Author |
|---|---|---|
| National Institutes of Health | RR12294, OD11024 | Margaret J McFall-Ngai, Edward G Ruby |
| National Institutes of Health | AI50661 | Margaret J McFall-Ngai |
| National Institutes of Health Molecular Biosciences Training Grant to University of Wisconsin–Madison | | Caitlin A Brennan |
| National Institutes of Health Microbes in Health and Disease Training Grant | | Caitlin A Brennan |

The funders had no role in study design, data collection and interpretation, or the decision to submit the work for publication.

## Author contributions

CAB, Conception and design, Acquisition of data, Analysis and interpretation of data, Drafting or revising the article; JRH, Acquisition of data, Analysis and interpretation of data; NK, BCK, MAA, Analysis and interpretation of data, Drafting or revising the article; MJM-N, EGR, Conception and design, Analysis and interpretation of data, Drafting or revising the article

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
