## [Decision Letter]

Thank you for sending your work entitled “Rotation of sheathed bacterial flagella promotes an immune response during host colonization in a model symbiosis” for consideration at *eLife*. Your article has been favorably evaluated by a Senior editor and 3 reviewers, one of whom is a member of our Board of Reviewing Editors.

The Reviewing editor and the other reviewers discussed their comments before we reached this decision, and the Reviewing editor has assembled the following comments to help you prepare a revised submission.

The three reviewers all believe your manuscript is very interesting and appropriate for *eLife*, but some modifications are needed for it to gain acceptance. This will require addition of some experimental data. The suggestions in the individual reviews are pretty clear and consistent:

1) We feel that you need to show that the levels of LPS in wildtype vs mutant supernatants can account for the relative capacities to induce apoptosis.

2) We also think it is important to measure peptidoglycan in the supernatants. Your previous work shows it is involved in the apoptotic response or some other way nailing down the conclusion that it is specifically shed LPS that is responsible for apoptosis.

We hope you can modify the manuscript in accordance with the reviewer suggestions below. It will then make a very nice contribution to the journal. Please provide a point-by-point disposition with a revised manuscript.

*Reviewer #1*:

Brennen et al. is a clear easy read that uncovers a very interesting function of the sheath encasing flagella of a handful of bacterial species that exhibit important associations with animals. These associations include those of important human pathogens like *Vibrio cholerae*. The work uncovered that rotation of the flagella results in release of LPS (endotoxin) from bacterial cells. The LPS serves to stimulate apoptosis in relevant host cells. What I particularly like about the manuscript is that the authors use a special mutualistic association that is very well suited to uncover this novel function of sheathed flagella. They go on to present some limited evidence that this could very well be relevant in the pathogen V. cholerae. I have one single concern of substance and a number of more cosmetic issues.

The substantive concern is what is the role of rotating flagella in establishment of the symbiosis? Maybe I missed it in the paper. Can MotAB and MotX mutants successfully colonize the squid even though they can’t trigger apoptosis? If so, why and what does this say about the importance of flagellar sheath shedding and apoptosis? If the infection by Mot mutants does not proceed normally, where does it stall out?

Minor comments:

“Despite the flagellar sheath, flagellin monomers are a prevalent component of the secretome of *V. fischeri* (Graber and Ruby, unpublished data)”: Is it acceptable to cite unpublished data in *eLife*?

“Assuming our hypothesis is accurate” is unfortunate wording. Science should assume hypotheses are incorrect and attempt to disprove them by experimentation, not the other way around. Change wording to something about “To further test the hypothesis...”

“However, pleiotropic effects of mutations in these distinct pathways (L. McCarter, personal communication) prevent direct comparison of unsheathed and sheathed flagella within one species”: Another personal communication.

“As this behavior is difficult to probe experimentally, novel approaches, such as the modeling of the biophysical properties underlying lipid interactions at a sub-micron scale, are needed to inform our understanding of the forces at play between a rotating flagellar sheath and the contiguous, but static, cellular outer membrane”: This text is about how one might study this further. It is a vague statement about biophysical approaches. It is not useful.

Figure 1: I don't quite get this figure. It is just the presence or absence of bacteria in these locations? If so it should be clear. If something can be said about abundance it would be useful.

Figure 1—figure supplement 2: The images are of full-page blow-ups of single bacterial cells (one with flagella and one without). First they don't need to be this big. Second, and more important, images of single cells are not convincing.

Figure 2—figure supplement 1: Bacterial growth is usually plotted on a log scale over linear time because growth is logarithmic. I recommend revising this figure.

*Reviewer #2*:

The manuscript by Ruby and colleagues presents an intriguing theory that the rotation of sheathed flagella found on certain bacteria results in release of lipopolysaccharide that can serve to stimulate host innate immune pathways. They present convincing data that *Vibrio fischeri* strains with no or amotile flagella produce less secreted LPS. They also present data showing that these strains have less capacity to induce epithelial apoptosis in the squid, the first step of symbiont-induced morphogenesis. They do not, however, establish whether the reduction in secreted LPS in these strains is responsible for the failure to induce morphogenesis. They also do not establish whether the alleged outer membrane vesicles at the tips of wild type flagella, observed in electron micrographs, require flagellar motility for their production.

1) A key basis for this paper is the assertion that LPS acts as the signal for the symbiont-induced morphogenesis of the squid host. The data in Figure 1—figure supplement 5 showing comparable apoptosis per ciliated field producing by lipid A exposure and *V. fischeri* colonization is different from the results reported in a previous paper from this group, by Koropatnick et al. 2004 (Figure 2), in which LPS has a modest effect inducing apoptosis and interacts synergistically with peptidoglycan to induce this effect. It's possible that lipid A is more potent than crude LPS. However, the real difference in these two sets of results is the extent of apoptosis induced by the symbiont, with significantly lower numbers in this manuscript's figure. The authors should comment on the differences between these two sets of results.

2) Based on the Koropatnick et al. 2004 results showing synergistic induction of apoptosis with combined LPS and PGN, it might be possible that the lack of morphogenesis induced by the flrA and motB mutants is due to the reduction of multiple morphogenic signals, as opposed to just LPS. For example, rotation of the flagellar shaft at the point of connection with the cellular membrane could enhance release of PGN. The authors should test whether levels of peptidoglycan and/or peptidoglycan monomers are reduced in the supernatant of the flrA, motB, and motX mutants relative to wild type cells. Additionally, the flagellin monomers that the authors state are secreted by wild type *V. fischeri* may act in conjunction with LPS to stimulate morphogenesis and these may well be reduced in the mutants.

3) The set of experiments that would convincingly show that the reduction in secreted LPS in the flrA and motB mutants is responsible for their failure to induce apoptosis would be to 1) show that secreted fractions from the wild type but not the mutant cells can induce apopotosis, and 2) show that depletion of LPS from the wild type cells (for example with the antibody used in Koropatnick et al.) eliminated that apoptosis-inducing activity in the secreted fraction from the wild type cells. These may be technically challenging experiments, but the authors need to provide more data on the connection between what these cells secrete and the functional capacities of these secretions in the symbiosis.

4) The authors discuss the issue of the threshhold concentrations of LPS to induce apoptosis, but they should provide more information on the functional and observed concentrations of these molecules. The experiment shown in Figure 1—figure supplement 5 uses 1 ug/ml lipid A. How does this compare to the concentrations of lipid A present in the LPS released by the flrA, motB, and motX mutants?

5) Based on the characterization of mutants with non-motile but intact flagella, the authors postulate that the biophysics of flagellar rotation result in LPS release. Their one piece of microscopic evidence for this is the electron micrograph in Figure 2, allegedly showing vesicular release from flagellar tips. The authors should provide more analysis of this phenomenon. First are the alleged vesicles comparable in size to outer membrane vesicles? Second, at what frequency are they observed? Finally, if they are the result of flagellar rotation, then they should not be observed in preparations of cells with amotile flagella. Is this the case?

*Reviewer #3*:

The manuscript by Brennan et al. investigates the sheathed flagellum of Vibrio fischeri. The reason for the sheath, an extension of the outermembrane, covering the filament has remained mysterious. The authors suggest that it may be provoking an immune response to LPS. They showed that LPS is shed into the environment when the flagellum is rotating, and that bacteria that cannot rotate their flagellum are defective at inducing apoptosis in host squid cells. They additionally showed that a flagellated but non-motile *V. cholerae* strain also releases less LPS into the environment and suggest that this might be conserved among all bacteria with sheathed flagella. In general the observation is very interesting and may point at a reason for the existence of the sheath surrounding the flagellar filament. Specific comments on the manuscript:

1) One piece was missing from their story: they showed that flagellated non-motile bacteria are defective at inducing apoptosis, and that these bacteria shed less LPS, but they didn’t show that the supernatants from these bacteria (vs the wildtype supe) are defective at inducing apoptosis. Given their argument that the shed LPS is what is stimulating apoptosis, they should show that the supes from the bacteria do/don’t induce apoptosis in the squid cells, rather than using purified LPS. They should do the same with the *V. cholerae* supes as well, since the source of the LPS shouldn’t make much difference.

2) Another control should be included, which is the addition of phenamil, a specific inhibitor of the sodium-driven motor, to the wildtype cells-this will stop flagellar rotation, and if their hypothesis is correct, there will be less LPS in the supe, which will stimulate less apoptosis.

---

## [Author Response]

*1) We feel that you need to show that the levels of LPS in wildtype vs mutant supernatants can account for the relative capacities to induce apoptosis*.

We understand the concern underlying this comment; however, there are two experimental realities that make this point not as easily resolvable as we, or the reviewers, would like. First, addition of culture medium is toxic to the animal, so this approach is untenable. Second, LPS is not a molecule amenable to simple solution-chemistry assumptions; i.e., it is produced by bacterial cultures in several chemical and physical forms (Phillips et al., 2011), each with a different specific activity, depending on the target/receptor.

For these reasons, which we elaborate upon below in the specific responses, our work simply proposes that because (i) non-rotating flagella release less total LPS, and (ii) LPS is the only molecule known to induce early-stage morphogenesis, then the most parsimonious and reasonable (though not airtight) explanation is that non-rotating flagella induce less early-stage morphogenesis because of their lower level of LPS release. We also feel that the strength and impact of the manuscript is in its use of the squid-vibrio system to discover a new phenomenon of general importance (see Reviewer #1’s first paragraph), rather than in using the phenomenon to describe something new about already extensively studied signals in this specific symbiosis.

To address the reviewers’ concern, we have changed the title, and made modifications throughout the manuscript, as well as others noted in the specific responses, below. These changes do not lower the impact of the work, which uses a model symbiosis to unveil a new biological behavior that occurs in *V. fischeri, V. cholerae,* and perhaps other bacteria, with consequences of importance in host-microbe interactions.

*2) We also think it is important to measure peptidoglycan in the supernatants. Your previous work shows it is involved in the apoptotic response or some other way nailing down the conclusion that it is specifically shed LPS that is responsible for apoptosis*.

We thank the reviewers for their careful consideration of the published literature in examining our work. However, there appears to be some confusion with regard to this point. LPS, particularly the lipid-A moiety, specifically induces early-stage apoptosis (as measured by acridine-orange staining) during symbiotic initiation (the appropriate reference for these data is [10], not Koropatnick et al., 2004). The Koropatnick et al. paper measures apoptosis after 24 h, well into the persistence stage of the symbiosis, and many hours after *V. fischeri* cells have already colonized the light-organ crypts and undergone substantial proliferation (∼100,000-fold). At such post-colonization time points, LPS/lipid A synergizes with TCT to drive apoptosis into its later stages. In contrast, at the 10-h time point examined in this study, LPS/lipid A is the inducing molecule, and TCT has no known synergistic effect. This difference in experimental timing (10 h vs 24 h) also accounts for the higher numbers of apoptotic nuclei reported in that publication (as noted by Reviewer 2).

Additionally, as indicated in the Methods, all of our experiments were performed in the presence of exogenously added peptidoglycan, which likely contains TCT as a breakdown product, a point that we have added to the text for clarification. Significantly, this cue is always present in natural seawater because of the presence of bacterioplankton, and it causes the hatchling squid to produce mucus, a requirement for efficient colonization (31). The level of peptidoglycan added to the bacteria-free artificial seawater to induce mucus shedding in this experiment would likely negate the effect of any differences in TCT release between the test strains and the aposymbiotic condition at this early time point (Figure 1, panels B and F). In fact, the absence of significant apoptosis in the aposymbiotic squid (which were exposed to the added peptidoglycan) confirms that this addition is insufficient to induce acridine-orange staining (Figure 1). We have added this point to the figure legend for clarification. Further, unpublished data from a different study that examined another host response during initiation (hemocyte trafficking into the light organ appendages, which is induced by TCT alone) indicates there is no significant difference in this TCT-dependent response between squid exposed to either wild-type or *motB1* strains of *V. fischeri*.

Nevertheless, we agree with the reviewer that it certainly remains possible that, in addition to LPS, other molecules (peptidoglycan, flagellin, outer membrane proteins, etc.) are differentially released upon flagellar rotation, and we have added this idea to the Discussion.

Reviewer #1:

*Brennen et al. is a clear easy read that uncovers a very interesting function of the sheath encasing flagella of a handful of bacterial species that exhibit important associations with animals. These associations include those of important human pathogens like Vibrio cholerae. The work uncovered that rotation of the flagella results in release of LPS (endotoxin) from bacterial cells. The LPS serves to stimulate apoptosis in relevant host cells. What I particularly like about the manuscript is that the authors use a special mutualistic association that is very well suited to uncover this novel function of sheathed flagella. They go on to present some limited evidence that this could very well be relevant in the pathogen V. cholerae. I have one single concern of substance and a number of more cosmetic issues*.

*The substantive concern is what is the role of rotating flagella in establishment of the symbiosis? Maybe I missed it in the paper. Can MotAB and MotX mutants successfully colonize the squid even though they can’t trigger apoptosis? If so, why and what does this say about the importance of flagellar sheath shedding and apoptosis? If the infection by Mot mutants does not proceed normally, where does it stall out*?

We thank the reviewer for alerting us to the need to be explicit about this point. We have added a phrase to clarify that amotile mutants, including the *motB1* and *motX* mutants, do not successfully colonize the juvenile light organ. The hypothesis in the field, as with other systems, is that it is the inability of these mutants to correctly localize to their symbiotic niche – in this case, the deep crypts – that prevents colonization, but the precise stage/location at which initiation is arrested has not yet been thoroughly determined. Therefore, it remains possible that the defects of the strains in colonizing the light organ are ultimately due, in part, to defects in host recognition/response rather than simply in their failure to migrate into the deep crypts. To clarify this latter point, we have added a phrase to the text.

*Minor comments*:

*“Despite the flagellar sheath, flagellin monomers are a prevalent component of the secretome of V. fischeri (Graber and Ruby, unpublished data)”: Is it acceptable to cite unpublished data in* eLife?

While the data from *V. fischeri* are unpublished, the results are not unanticipated from studies of other sheathed bacteria. We have maintained the citation to our unpublished data, but we have also added a reference to work performed in *V. cholerae* (40) and modified the sentence to strengthen our statement.

*“Assuming our hypothesis is accurate” is unfortunate wording. Science should assume hypotheses are incorrect and attempt to disprove them by experimentation, not the other way around. Change wording to something about “To further test the hypothesis...*”

We agree with the reviewer, and we have modified this sentence in accord with their comment.

*“However, pleiotropic effects of mutations in these distinct pathways (L. McCarter, personal communication) prevent direct comparison of unsheathed and sheathed flagella within one species”: Another personal communication*.

While we would like to replace this personal communication with an appropriate citation, the problem we are addressing has not been explicitly stated in the literature. Linda McCarter is the expert in the field of *Vibrio parahaemolyticus* flagellar motility, and, when asked about the potential for such an experiment, she responded “V[ibrio] P[arahaemolyticus] is complicated by the biology of the two interacting flagellar systems and the not well-matched backgrounds of the existing mutants,” and described the collateral effects upon other cellular behaviors of disrupting either flagellar system. (Dr. McCarter gave us permission to quote her in our response.) We have modified this sentence to more accurately reflect her observation.

*“As this behavior is difficult to probe experimentally, novel approaches, such as the modeling of the biophysical properties underlying lipid interactions at a sub-micron scale, are needed to inform our understanding of the forces at play between a rotating flagellar sheath and the contiguous, but static, cellular outer membrane”: This text is about how one might study this further. It is a vague statement about biophysical approaches. It is not useful*.

We would like to argue that this text helps place our research into a broader context. In numerous discussions of the flagellar sheath with other scientists, including both biophysicists and microbiologists who are experts on the outer membrane and its stresses, it has remained a point of amazement that no one has considered how such a modification to the flagellar apparatus might affect membrane stability. Therefore, we request permission to retain this single sentence in the Discussion so that readers are made aware of this current lack of understanding about sheathed-flagellum function.

Figure 1*: I don't quite get this figure. It is just the presence or absence of bacteria in these locations? If so it should be clear. If something can be said about abundance it would be useful*.

The data in Figure 1 demonstrate that the tissue location of the *flrA* and *motB1* mutants is the same as that of wild-type cells at the time we are comparing them (i.e., the earliest stages of initiation), and that the phenotypes we report for these mutants do not result simply from a differential arrest earlier in migration into the light organ crypts (as mentioned above, the point of their ultimate arrest has not yet been defined). However, we can appreciate the reviewer’s confusion with this figure, and have added a sentence in the Materials and methods to clarify that these data indicate the relative tissue location of mutant and wild-type bacteria at the time of our assays.

Concerning abundance data, while the squid-vibrio system allows much greater resolution of the initiation process than vertebrate models, enumerating bacteria at different locations within live animals still is a difficult task, especially after the bacteria get further into host tissue (e.g., through the pores and into the ducts and antechamber). However, as indicated in the figure legend, we have paired the data in Figure 1 with a determination of the total colony-forming units in the entire organ (Figure 1), and these values are indistinguishable between the strains.

Figure 1—figure supplement 2*: The images are of full-page blow-ups of single bacterial cells (one with flagella and one without). First they don't need to be this big. Second, and more important, images of single cells are not convincing*.

We apologize for the confusion. These images are not intended to be full-sized. We believe that’s just the way the upload-process handles the figure supplements. The images are provided to support the statements that the *flrA* and *motB1* mutants have the expected flagellar phenotypes (aflagellate for *flrA*, and flagellate for *motB1*) just like in homologous mutants in other bacteria. The images are simply representative cells, which we have clarified in the figure titles (Figure 1—figure supplement 2 and Figure 1—figure supplement 3, and Figure 4—figure supplement 2). While expected, because of studies of homologous mutants in *V. cholerae* and other *Vibrio spp.*, the presence or absence of a flagellum had not been described previously for neither any known flagellar motor mutant in *V. fischeri* nor this particular *flrA* mutant, although these data have been reported for a previous construction of an *flrA* mutant (28). As these results are unsurprising, such confirmation seems useful, but not essential, for interpretation of our results, and is placed as supplementary information.

Figure 2—figure supplement 1*: Bacterial growth is usually plotted on a log scale over linear time because growth is logarithmic. I recommend revising this figure*.

We have modified the figure per the recommendation of the reviewer.

Reviewer #2:

*The manuscript by Ruby and colleagues presents an intriguing theory that the rotation of sheathed flagella found on certain bacteria results in release of lipopolysaccharide that can serve to stimulate host innate immune pathways. They present convincing data that Vibrio fischeri strains with no or amotile flagella produce less secreted LPS. They also present data showing that these strains have less capacity to induce epithelial apoptosis in the squid, the first step of symbiont-induced morphogenesis. They do not, however, establish whether the reduction in secreted LPS in these strains is responsible for the failure to induce morphogenesis. They also do not establish whether the alleged outer membrane vesicles at the tips of wild type flagella, observed in electron micrographs, require flagellar motility for their production*.

The reviewer’s comment indicates a desire to directly relate the amount of LPS released by colonizing bacteria to an amount of added purified LPS sufficient to induce morphogenesis (in the absence of bacteria). However, as mentioned in our first response, above, as well as in new text, working with LPS is far more complicated than typical solution chemistry. The addition of exogenous LPS assumes no context-dependency in terms of either bacterial proximity (e.g., the binding of *V. fischeri* to the host cells/receptors) or the form of LPS (micelles, vesicles, fragment size, etc.) presented to the host. As we mention in the Discussion, we expect that the manner in which LPS is presented is an important part of the way it is sensed by the host cells, and that culture supernatants contain LPS released not only by flagellar rotation, but also from other sources like outer membrane vesicles, cell lysis, etc, that have accumulated during growth. In the context of symbiotic initiation, during which there is no *V. fischeri* proliferation within the timeframe of our experiments, such other sources are less likely to be relevant to the induction of early-stage apoptosis. Taken together, we believe that the proposed experiment would be difficult to interpret, problematic to quantify, and provide little biologically relevant insight into the host responses that are naturally triggered by colonizing *V. fischeri* cells.

*1) A key basis for this paper is the assertion that LPS acts as the signal for the symbiont-induced morphogenesis of the squid host. The data in*
Figure 1*–figure supplement 5 showing comparable apoptosis per ciliated field producing by lipid A exposure and* V. fischeri *colonization is different from the results reported in a previous paper from this group, by Koropatnick et al. 2004 (*Figure 2*), in which LPS has a modest effect inducing apoptosis and interacts synergistically with peptidoglycan to induce this effect. It's possible that lipid A is more potent than crude LPS. However, the real difference in these two sets of results is the extent of apoptosis induced by the symbiont, with significantly lower numbers in this manuscript's figure. The authors should comment on the differences between these two sets of results*.

We thank the reviewer for their thorough reading of past literature in considering this manuscript. As described in an earlier response (combined-review comment #2), the different numbers of apoptotic nuclei reported in the two works is because they are examining the tissues at two different time points during symbiotic colonization. In Koropatnick et al., the authors were examining a later time point (24 h post-inoculation), by which time the symbionts have colonized the light organ’s deep crypts and undergone many rounds of doubling. In the work presented here, we are examining the induction of early-stage apoptosis that occurs when non-growing *V. fischeri* cells are still initiating the symbiosis (10 h post-inoculation), well before they reach the deep crypts, where subsequent proliferation occurs. While inoculated animals have significantly higher apoptosis levels than uninoculated ones at both 10 and 24 h, this developmental (and experimental) difference accounts for the higher absolute numbers of apoptotic nuclei observed in the Koropatnick et al. study.

*2) Based on the Koropatnick et al. 2004 results showing synergistic induction of apoptosis with combined LPS and PGN, it might be possible that the lack of morphogenesis induced by the flrA and motB mutants is due to the reduction of multiple morphogenic signals, as opposed to just LPS. For example, rotation of the flagellar shaft at the point of connection with the cellular membrane could enhance release of PGN. The authors should test whether levels of peptidoglycan and/or peptidoglycan monomers are reduced in the supernatant of the flrA, motB, and motX mutants relative to wild type cells. Additionally, the flagellin monomers that the authors state are secreted by wild type* V. fischeri *may act in conjunction with LPS to stimulate morphogenesis and these may well be reduced in the mutants*.

We have addressed the PGN/TCT issue in the response to the combined review, above. As for the cellular location of LPS shedding, we completely agree that another possible source of the released LPS could be the interface at the base of the flagellum, between the flagellar sheath and the static outer membrane. Currently, though, there exists no evidence to support this hypothesis. It would clearly be an important next step to ask whether there are additional molecules (such as flagellin) that might be differentially shed between these strains, and what their activity might be (synergistically or otherwise). However, in this manuscript we have described a new role for the rotation of the sheathed flagellum, and believe that testing all other candidate surface components for their presence and possible synergy will take considerable effort, and is beyond the scope of this initial announcement and characterization of an important discovery.

*3) The set of experiments that would convincingly show that the reduction in secreted LPS in the flrA and motB mutants is responsible for their failure to induce apoptosis would be to 1) show that secreted fractions from the wild type but not the mutant cells can induce apopotosis, and 2) show that depletion of LPS from the wild type cells (for example with the antibody used in Koropatnick et al.) eliminated that apoptosis-inducing activity in the secreted fraction from the wild type cells. These may be technically challenging experiments, but the authors need to provide more data on the connection between what these cells secrete and the functional capacities of these secretions in the symbiosis*.

We are unable to perform the first suggested experiment because, as mentioned above, we can’t add culture supernatants to seawater because they are toxic to the animals; too many other metabolic waste products accumulate during the culture’s growth. One could argue that we can simply isolate the LPS from those supernatants, and add it to the animal’s seawater, but that is essentially what we have done already with the addition of purified LPS. Also, as noted earlier, choosing what amount of this material to add would be arbitrary, and impossible to quantitatively relate to the natural process, which is initiated by a few non-growing cells attached to host tissue in a active flow field.

For the same reason we can’t remove the LPS from these supernatants with LPS antibody, as the toxic products will still be present. In addition, our previous published studies show that, while treating the seawater with antibody removes LPS, such an addition can not be made during a colonization experiment, presumably because the antibody binds the *V. fischeri* cells (including the flagellar sheath) and interferes with the bacterium’s surface association with the host cilia. This is not even a specific effect: as a similar blocking of colonization occurs with the addition of an antibody to OmpU, the bacterium’s major outer membrane porin (Aeckersburg et al., 2001).

*4) The authors discuss the issue of the threshhold concentrations of LPS to induce apoptosis, but they should provide more information on the functional and observed concentrations of these molecules. The experiment shown in*
Figure 1*–figure supplement 5 uses 1 ug/ml lipid A. How does this compare to the concentrations of lipid A present in the LPS released by the flrA, motB, and motX mutants*?

As we described in response to a previous comment, the concept of a ‘concentration’ of LPS present in a culture is problematic, and implies that all the possible forms LPS comes in have the same specific activities for a given response. Our purpose in mentioning a threshold LPS level is only to suggest that a relatively small difference in LPS release might trigger a significant host response. Because the light organ’s surface epithelium is constantly exposed to seawater, it experiences the low levels of LPS generated by the presence of the naturally occurring bacterioplankton; thus, these host cells must differentiate between an ‘authentic’ indicator of the presence of the symbiont (∼10^3^ cells/ml), and this background ‘noise’ of the normal bacterioplankton (>10^6^ cells/ml). Such a dilemma suggests that the signaling pathways are finely tuned to the additional LPS presented by the rotating flagella of the 5-10 *V. fischeri* cells attached directly to the host tissue (Altura et al). Thus, it is hard to estimate the amount of LPS released by individual cells during symbiotic initiation. As previously mentioned, the few *V. fischeri* cells initiating the response are not growing during the early hours of the association examined here, and the dilution effect of the flow field around the cilia to which they are attached is a mystery; thus, even back-of-the envelope calculations based on LPS release in culture require substantial, unsupported, assumptions be made. In any case, we agree that this is an important point to clarify for the reader, and have added to the text to describe it.

The data (Figure 1—figure supplement 4) the reviewer mentions serve merely as a control for our experiments, confirming that, as in [10], lipid A is sufficient to induce early-stage apoptosis under the conditions used in our work. Given that this experiment is only a methodological control, and not critical to our argument, we are willing to remove these data if the editor feels they distract the reader from the overall message of the manuscript.

*5) Based on the characterization of mutants with non-motile but intact flagella, the authors postulate that the biophysics of flagellar rotation result in LPS release. Their one piece of microscopic evidence for this is the electron micrograph in*
Figure 2*, allegedly showing vesicular release from flagellar tips. The authors should provide more analysis of this phenomenon. First are the alleged vesicles comparable in size to outer membrane vesicles? Second, at what frequency are they observed? Finally, if they are the result of flagellar rotation, then they should not be observed in preparations of cells with amotile flagella. Is this the case*?

We can see that our original wording was not clear, and we have clarified our intentions in providing this image. Rather than suggesting that these vesicle-like structures are the definitive source of shed LPS (which, as the reviewer points out, would require answering a number of other descriptive and experimental questions), we intended these data to support the notion that structural weaknesses can appear between the flagellar sheath and shaft, analogous to the weaknesses between the outer membrane and peptidoglycan layer required for cellular OMV formation. Further, their location is consistent with similar structures observed in the sheathed flagella of *Helicobacter pylori* (Geis et al.; Josenhans et al.).

We have not performed quantitative analyses of vesicle formation from EMs because one could argue that it would be (i) less likely to see them on the non-rotating mutants, as they would not form as rapidly, or (ii) more likely to see them, as, once formed, they would not be as easily shed. Instead, we feel the most rigorous and quantitative indication of LPS shedding is the measurement of LPS in culture supernatants we performed. Thus, we have now made it clear (line 115) that the illustration of vesicle-like structures associated with the flagellum is provided simply to indicate a possible sheath-related source of the LPS appearing in the supernatant.

As discussed later, it is quite possible that the release of LPS more typically occurs elsewhere, such as at the base of the flagellum. Parsing out the location(s) of LPS release, and their chemical form(s), is the subject of another ongoing study requiring the development of new techniques that are beyond the scope of this paper. Here, our goal is to provide new data on a subject that has made very little advancement over the past 30 years, and to share our findings at this stage of our studies so as to encourage a reopening of flagellar-sheath research by investigators using different bacterial species, any one of which may prove to be a key model for future discoveries.

Reviewer #3:

*The manuscript by Brennan et al. investigates the sheathed flagellum of Vibrio fischeri. The reason for the sheath, an extension of the outermembrane, covering the filament has remained mysterious. The authors suggest that it may be provoking an immune response to LPS. They showed that LPS is shed into the environment when the flagellum is rotating, and that bacteria that cannot rotate their flagellum are defective at inducing apoptosis in host squid cells. They additionally showed that a flagellated but non-motile V. cholerae strain also releases less LPS into the environment and suggest that this might be conserved among all bacteria with sheathed flagella. In general the observation is very interesting and may point at a reason for the existence of the sheath surrounding the flagellar filament. Specific comments on the manuscript*:

*1) One piece was missing from their story: they showed that flagellated non-motile bacteria are defective at inducing apoptosis, and that these bacteria shed less LPS, but they didn’t show that the supernatants from these bacteria (vs the wildtype supe) are defective at inducing apoptosis. Given their argument that the shed LPS is what is stimulating apoptosis, they should show that the supes from the bacteria do/don’t induce apoptosis in the squid cells, rather than using purified LPS. They should do the same with the V. cholerae supes as well, since the source of the LPS shouldn’t make much difference*.

As described above, addition of bacterial culture supernatants to live animals is not a viable approach.

*2) Another control should be included, which is the addition of phenamil, a specific inhibitor of the sodium-driven motor, to the wildtype cells-this will stop flagellar rotation, and if their hypothesis is correct, there will be less LPS in the supe, which will stimulate less apoptosis*.

We appreciate and agree with the logic underlying this reviewer’s experiment and, not surprisingly, we had already examined the effect of phenamil on LPS release in these strains, producing the data shown here (see Figure 6). In comparing cell-free culture supernatants exposed to either DMSO (the solvent control) or phenamil, we observed that, upon addition of phenamil, the *motB1* mutant no longer exhibited reduced LPS shedding as compared to wild-type cells similarly exposed to phenamil, supporting the conclusion that both chemical disruption of the sodium-motor pump and flagellar rotation serve the same functions in LPS shedding. However, the overall level of LPS in wild-type supernatants was not reduced after exposure to phenamil, but, rather, it trends toward being slightly higher for both strains, compared to the DMSO controls.Author response image 1.

A possible explanation for this difference became apparent when we prepared the cells for this experiment; phenamil addition delayed the growth of *V. fischeri*, even at concentrations at which cells are still motile when observed by microscopy. Thus, addition of phenamil to the culture supernatants does not specifically interrupt flagellar rotation, as hoped, but rather has collateral effects on *V. fischeri* physiology and/or stress response*.* For instance, as a marine organism, *V. fischeri* uses sodium pumps not only for its flagellar motor, but also for nutrient transport via sodium symporters. With this caveat in mind, these data could be interpreted as supporting our hypothesis. However, we cannot rule out that these indirect effects of phenamil exposure might mask other differences. While we did not include these phenamil data in the final publication, because we were concerned they might be misinterpreted by the reader, we are certainly willing to do so at the request of the editor.